# Treatment of Spent Pickling Solutions by Diffusion Dialysis Using Anion-Exchange Membrane Neosepta-AFN

**DOI:** 10.3390/membranes13010009

**Published:** 2022-12-21

**Authors:** Helena Bendová, Libor Dušek

**Affiliations:** Institute of Environmental and Chemical Engineering, Faculty of Chemical Technology, University of Pardubice, 53210 Pardubice, Czech Republic

**Keywords:** continuous diffusion dialysis, hydrofluoric acid, ferric nitrate, spent pickling solution, anion-exchange membrane

## Abstract

This article presents the possibility of using diffusion dialysis for processing spent pickling solution from pickling stainless steels with a mixture of nitric acid and hydrofluoric acid. A counter-current two-compartment dialyzer equipped with an anion-exchange membrane Neosepta-AFN was used to study and compare the diffusion dialysis of model mixture of hydrofluoric acid and ferric nitrate and a real spent pickling solution. The separation efficiency was characterized by the acid recovery yield, the rejection coefficient of the metals, the permeability coefficient of the membrane, and the separation factor. These characteristics were calculated from the data obtained at steady state. For the real spent pickling solution tested, the permeability values of nitrates 1.7 × 10^−6^ m s^−1^, fluorides 0.4 × 10^−6^ m s^−1^, and ferric ions 1.1 × 10^−7^ m s^−1^ were achieved. The separation factor for nitrates/ferric ions was 15.7 and 3.6 for fluorides/ferric ions. Furthermore, the dependencies of recovery yield and rejection for different concentrations of hydrofluoric acid and ferric nitrate were determined.

## 1. Introduction

During industrial pickling of metal surfaces, a large number of waste pickling baths are produced throughout the world [1,2]. These spent pickling solutions have a relatively high content of free acids and metal salts formed by the dissolution of the surface layers. The type and concentration of acids, as well as dissolved salts, are highly dependent on the type of pickling bath and its use [1]. Hydrochloric acid and sulfuric acid are used for pickling ferritic and high-speed steels, and mixtures of nitric acid with hydrofluoric acid are used for pickling austenitic steels and alloys. A spent pickling solution (SPS) cannot be used any further, because the efficiency of the pickling process decreases with increasing content of dissolved metals in the bath. They are classified as hazardous waste, with a negative impact on the environment. Conventional methods for processing of SPS include their elimination by neutralization and the subsequent disposal method, which is no longer considered the Best Available Technique (BAT) [2]. Technologies that allow at least partial regeneration of SPS are evaporation, crystallization, use of ion exchangers, pyrometallurgical methods, metal extraction using suitable solvents and membrane separation processes (diffusion dialysis, electrodialysis, membrane distillation, and electrodialysis with bipolar membranes) [1].

Diffusion dialysis, which belongs to a group of promising separation processes, is preferably used to recover inorganic acids from acid waste waters generated in the steel, metal-refining, and electroplating industries. It is a spontaneous process, with a driving force that is the concentration difference between two solutions separated by an ion-exchange membrane. Its main advantages include a high acid recovery yield, high rejection of metals, low environmental impact, and low energy consumption. The only energy is that of ensuring the transport of liquid streams into a dialyzer. However, diffusion dialysis is a very slow process because its controlling step is the transport of components through the membrane by diffusion.

In recent years, there have been several studies on the regeneration of waste solutions by diffusion dialysis [3,4,5,6,7,8,9,10,11,12,13,14,15,16,17,18,19]. Lan et al. [3] published results on the recovery of HNO_3_ from acid effluents discharged by an electrodialysis plant. Diffusion dialysis with anion-exchange membranes was used to recover H_2_SO_4_ from an acid leaching solution generated in the vanadium-producing process [4]. Kim et al. [5] studied the recovery of phosphoric acid from mixed waste acids from the semiconductor industry (containing acetic acid, nitric acid, and phosphoric acid with aluminum and/or molybdenum) by diffusion dialysis and vacuum distillation. In the literature [6], spent solder-stripping solutions with the content of tin, copper, iron, and lead in nitric acid were treated by diffusion dialysis after thermal precipitation of tin. Xiao et al. [7] studied the selective removal of halides from the spent zinc sulfate electrolyte by continuous diffusion dialysis. Wang et al. [8] used diffusion dialysis to recover sulfuric acid from a stone coal acid leaching solution. Amrane et al. [9] coupled diffusion dialysis with precipitation cementation to separate and recover nitric acid, Cu^2+^, Zn^2+^ and Pb^2+^ from wastewater from a brass pickling bath. Bendova and Weidlich [10] presented the application of continuous diffusion dialysis using anion-exchange membrane Neosepta-AFN in the hydrometallurgical separation of nickel from the spent Raney Ni catalyst.

Gueccia et al. [11] used diffusion dialysis for the separation of hydrochloric acid from iron and zinc from highly concentrated pickling solutions using a laboratory batch dialyzer and a larger continuous dialyzer, both equipped with a Fumasep anion-exchange membrane. In their subsequent work, an innovative membrane process was proposed for acid recovery from the pickling solution by combining diffusion dialysis and membrane distillation technologies with a reactive precipitation unit [12], a pilot operation of this combined process was presented [13], and an economic analysis was performed to demonstrate the feasibility of this developed process [14]. To overcome the limitations of the traditional DD process, such as low processing capacity and serious water osmosis, Zhang et al. [15] proposed a double-driven pressure–concentration DD process. Du et al. [16] studied the selective removal of chloride from the adipate formation bath in the foil industry by diffusion dialysis. The recovery of rare earth elements from electronic waste by diffusion dialysis was investigated by the authors [17]. In the latest study [18], the problem of sulfuric acid recycling from spent copper plating solution (containing H_2_SO_4_, FeSO_4_, and CuSO_4_) was solved using a hybrid membrane technology. Diffusion dialysis was used to separate sulfuric acid and salts of heavy metals and then purified dilute sulfuric acid was concentrated by electrodialysis.

To determine the characteristics of the DD process, two types of equipment are used, a batch cell and a continuous dialyzer. In the first case, the data on time dependences of the liquid volumes and component concentrations in the individual compartments are treated, while in the latter case, the characteristics are calculated from the volumetric liquid flow rates and concentrations of all streams at steady state. In the continuous process, the flat plate dialyzer is the most commonly used type of membrane module; however, the space saving characteristics and modular nature of spiral wound DD membrane modules have recently attracted much attention [19,20]. The capacity of a spiral-wound module equipped with the Fumasep-FAD membrane designed for continuous recovery of free acid using DD to separate sulfuric acid and Cu^2+^ and Fe^2+^ salts [19] and hydrochloric acid and Zn^2+^, Ni^2+^, Cr^3+^, and Fe^2+^ salts [20] was studied.

The aim of this study is to compare the continuous diffusion dialysis of a model mixture of hydrofluoric acid and ferric nitrate and a real spent steel pickling solution and establish the basic characteristics of this DD process. Model mixtures were selected to approximate the composition of the liquor from the pickling of stainless steels in the mixture of nitric acid and hydrofluoric acid.

## 2. Theory

Consider a continuous dialyzer with two identical compartments with an anion-exchange membrane. As shown in Figure 1, the feed enters the bottom of compartment *I*, while the stripping agent (water) enters the top of compartment *II*. To quantify the diffusion dialysis of liquid mixtures at steady state, the recovery yield, *ν_i_*, and the rejection coefficient, *R_i_*, are used
(1)$$\nu_{i}=\frac{{\dot{V}}_{out}^{II}c_{i,out}^{II}}{{\dot{V}}_{in}^{I}c_{i,in}^{I}}\times 100\%$$
(2)$$R_{i}=\left(1-\frac{{\dot{V}}_{out}^{II}c_{i,out}^{II}}{{\dot{V}}_{in}^{I}c_{i,in}^{I}}\right)\times 100\%$$

Equation (1) was used to calculate the acid recovery yield (HF + HNO_3_), and Equation (2) to determine the rejection coefficient for metal ions (Fe^3+^, Cr^3+^ and Ni^2+^).

From the balance of *i* ions over the differential volume of compartments *I* and *II* written in steady state, using the definition of derivation and the following arrangement [21,22], it is possible to obtain the differential equations describing the concentration dependence of *i* ions on the length coordinate *z*
(3)$$\frac{dc_{i}^{j}}{dz}\;=\;-\frac{1}{{\dot{V}}^{j}}\frac{A}{z_{T}}J_{i}-\frac{c_{i}^{j}}{{\dot{V}}^{j}}\frac{d{\dot{V}}^{j}}{dz}$$

$\dot{V}$ is the volumetric liquid flow rate, *c* is the molar concentration, *J* is the molar flux, *A* is the membrane area, and *z_T_* is the height of the compartment. Superscripts *j* = *I*, *II* mean the compartments.

If the concentrations of *i* ions and volumetric liquid flow rates of all the streams connected to the dialyzer are known in steady state, then it is possible to numerically integrate the set of differential Equation (3). If this step is followed by the suitable optimization procedure, the basic transport characteristic of the membrane/solution system can be obtained, i.e., the permeability of the membrane to *i* ions.

The flux of *i* ions through the membrane and through the liquid films can be expressed as
(4)$$J_{i}\;=\;k_{L}^{I}\left(c_{i}^{I}\;-\;c_{i,f}^{I}\right)$$
(5)$$J_{i}\;=\;P_{i}\left(c_{i,f}^{I}-\;c_{i,f}^{II}\right)$$
(6)$$J_{i}\;=\;k_{L}^{II}\left(c_{i,f}^{II}\;-\;c_{i}^{II}\right)$$

The subscript *f* means the solution/membrane interface, *P_i_* is the permeability of the membrane, and *k_L_* are the mass transfer coefficients, which can be determined from Equation (7) valid for laminar flow of liquid [23]
(7)Sh = C Re0.5Sc0.33

In this equation, *Sh* is the Sherwood number, *C* is a constant, *Re* is the Reynolds number, and *Sc* is the Schmidt number.

## 3. Materials and Methods

The diffusion dialysis was studied in a continuous flat plate two-compartment counter-current dialyzer, with compartments that were separated by an anion-exchange membrane Neosepta-AFN. This strongly basic polymeric membrane made from styrene and divinylbenzene is produced by Astom Corp. (Tokyo, Japan) and is specially developed for acid recovery from solutions containing acids and salts. Its basic properties determined experimentally were as follows: the thickness 160 μm, the water content 0.412 g per gram of dry membrane in Cl^−^ form and the concentration of fixed charges referred to the volume of a swollen membrane 1.74 mol L^−1^. The pre-treatment of the membrane was performed before the start of all the experiments. The membrane was transferred from chloride to nitrate and fluoride form by filling the dialyzer with the mixture of 0.1 mol L^−1^ HNO_3_ and 0.1 mol L^−1^ HF for 24 h. Subsequently, the dialyzer was washed thoroughly with water.

The height of the dialyzer was 1 m, the dimensions of each compartment were 0.92 m × 0.036 m × 0.0011 m, and the volume of each compartment 3.6 × 10^−5^ m^3^. The membrane area was 331 cm^2^. The dialyzer was placed in a box made of plexiglass, where the temperature was kept constant at 25 ± 0.5 °C. The feed entered the bottom of compartment *I*, while the distilled water entered the top of compartment *II*, the flows were ensured by peristaltic pumps. The achievement of the steady state (a time period from 2 to 5 h in the dependence on liquid flow rate), was indicated by constant concentrations of the components in three successive samples taken from the same stream. Then, the volumetric flow rates of all streams and the concentrations of ions in the outlet streams (dialysate and diffusate) were determined. The scheme of the experimental set-up can be found elsewhere [22].

The volumetric liquid flow rate of the inlet streams was from 1.4 × 10^−8^ to 7.0 × 10^−8^ m^3^ s^−1^ (50 to 250 mL h^−1^), and that of the feed was approximately equal to that of water. The flow rate relative to the membrane area was from 1.5 to 7.6 L h^−1^ m^−2^.

The initial concentration of hydrofluoric acid in the model mixture was 3.0 mol L^−1^, while that of ferric nitrate 0.7 mol L^−1^. In addition to the dialysis experiments with model mixtures, also experiments with the real spent pickling solution (provided by the company EKOMOR, s.r.o, Frýdek-Místek, Czech Republic) were carried out. The composition of the solutions tested is shown in Table 1 and compared with the literature [2]. The real spent pickling solution contains HF and Fe(NO_3_)_3_ as well as a small amount of Cr^3+^ and Ni^2+^ ions. Furthermore, comparative measurements were performed at one volumetric liquid flow rate (150 mL h^−1^), where the initial hydrofluoric acid concentrations were changed from 2.0 to 4.0 mol L^−1^, while that of ferric nitrate were changed from 0.5 to 0.9 mol L^−1^.

The concentration of metals (Fe, Cr and Ni) was determined by inductively coupled plasma optical emission spectroscopy (Integra 6000 ICP-OES, GBC Scientific Equipment, Dandenong, Victoria, Australia), the concentration of nitrate and fluoride by the electrophoretic analyzer EA 102, Villa Labeco s.r.o, Spišská Nová Ves, Slovakia (capillary zone electrophoresis) and that of H^+^ ions by titration with a standard NaOH solution or by flow coulometry (EcaFlow GLP150, Istran s.r.o., Bratislava, Slovakia).

## 4. Results and Discussion

### 4.1. Comparison of the Model Solution and the Spent Pickling Solution

In Table 2, the composition of dialysate and diffusate for the model mixture (3 M HF + 0.7 M Fe(NO_3_)_3_) and the real spent pickling solution are shown for diffusion dialysis with a volumetric liquid flow rate of 150 mL h^−1^ (the flow rate relative to the membrane area of 4.5 L h^−1^ m^−2^). Only ion concentrations were determined in the solutions, because it is not possible to determine the actual concentration values of the individual acids (HF and HNO_3_) in these mixtures that also contain metal ions. Therefore, the basic characteristics of the DD process (recovery, rejection, and permeability) are also determined for individual ions and not for concrete acids and salts (as is usual). The concentration of nitrates in the real spent solution tested was slightly higher than in the model mixture, while the concentration of fluorides was slightly lower, so the composition of the dialysate and the diffusate obtained also corresponded to this.

#### 4.1.1. Recovery Yield

The recovery yield values of individual ions (nitrates, fluorides, and hydrons) were calculated according to Equation (1) and summarized in Table 3 for the model mixture (MM) and the real spent pickling solution (SPS). The dependences of the ion recovery on the volumetric liquid flow rate are presented in Figure 2.

From Figure 2, it is evident that the recovery yield is strongly affected by the volumetric liquid flow rate. A decrease in the recovery yield can be seen with an increasing volumetric liquid flow rate. This is a result of a decrease in the mean residence time of the liquid in each compartment of the dialyzer. The recovery yield values of both nitrates and fluorides for the model mixture are slightly higher than those for the spent pickling solution, which is probably due to the different concentration of the solutions, and it will be explained below. The recovery yield values of nitrates are approximately 2.5-fold higher than those of fluorides at the same volumetric flow rate.

#### 4.1.2. Rejection Coefficient

The values of the rejection coefficient for individual metal ions were calculated according to Equation (2) and also summarized in Table 3. The dependences of the rejection coefficient for metal ions on the volumetric liquid flow rate are shown in Figure 3 for the model mixture (3 M HF + 0.7 M Fe(NO_3_)_3_) and the real spent pickling solution.

As expected, the rejection coefficients of metal ions increase significantly with increasing volumetric liquid flow rate, which is caused by a decrease in the mean residence time of the liquid in the dialyzer. For the model mixture, the rejection coefficients of Fe^3+^ ions were from 89 % to 97%. Approximately the same values of the rejection coefficient of Fe^3+^ ions and Cr^3+^ ions were measured for the spent pickling solution and only slightly lower values were determined for Ni^2+^ ions.

The dependencies of acid recovery yield and metal rejection upon volumetric liquid flow rate go ‘against each other’; therefore, it is necessary to choose the optimal value of the volumetric liquid flow rate for the given application and the desired results of diffusion dialysis. In this investigation, the volumetric liquid flow of 150 mL h^−1^ (the flow rate relative to the membrane area of 4.5 L h^−1^ m^−2^) was chosen as optimal, and the recovery and rejection dependencies on different acid and salt concentration were further determined for this flow rate value.

#### 4.1.3. Permeability of Membrane and Separation Factor

The permeability of the membrane was determined by numerical integration of the set of Equation (3), where *Ji* is expressed by Equation (5). The Runge–Kutta 4th-order method was used with the integration step *h* = 0.001 m. The set of Equation (3) was integrated in both directions of the length coordinate *z*. At each integration step, it was necessary to calculate the concentration of acid in liquid at the solution/membrane interface by solving Equations (4)–(6). Using an optimization procedure (Golden Section Search), such values of *Pi* were searched, at which the objective function calculated as the sum of the squared deviations of the experimental and calculated concentration values in all streams reached a minimum. The procedure to obtain the membrane permeability requires the constant *C* in Equation (7). Its determination is based on the consideration that the membrane permeability is a membrane/solution parameter, which is not affected by the flow of liquid, so that the values of permeability obtained at various liquid flow rates must not differ much. For this reason, a criterion *S* was defined as the sum of variances of *Pi* at a constant $c_{i,in}^{I}$, and its minimum was searched (for details, see [21,24]).

The separation factor was calculated as the ratio of ion permeabilities. The values of the permeability coefficients and separation factors are summarized in Table 4.

It can be seen from the table that the permeability for ferric ions is approximately one order of magnitude lower than that for the nitrates. Additionally, values of separation factor for nitrates/ferric ions reach relative high values. Anion-exchange membrane Neosepta-AFN proved to be good separator for this mixture of ions. Separation factors of fluorides/ferric ions is approximately 4-fold lower; the separation of these ions is therefore slightly worse. In addition, the permeability of nitrates is almost 4-fold higher than the permeability of fluorides, which corresponds to the previous results for ion recovery yield. The reason is that in the mixture of HNO_3_, HF, a ferric salt, ferric fluorocomplexes are formed according to equations
HF + Fe(NO_3_)_3_ ⇔ HNO_3_ + FeF(NO_3_)_2_HF + FeF(NO_3_)_2_ ⇔ HNO_3_ + FeF_2_NO_3_HF + FeF_2_NO_3_ ⇔ HNO_3_ + FeF_3_(8)

In the mixture, the FeF^2+^complex predominates and these divalent cations practically do not pass through the anion-exchange membrane. Therefore, HNO_3_ passes through the membrane faster than HF acid. In the course of DD, the balance is constantly disturbed in favor of further formation of fluorocomplexes and free HNO_3_ at the expense of free HF. In some cases, the amount of HNO_3_ in the diffusate can be greater than in the feed.

The values of permeability and separation factors for model mixture are slightly higher than those for real spent pickling solution, which again correspond to the previous results for ion recovery yield, and this is probably due to the slightly different composition of these mixtures. Nevertheless, it can be stated that using continuous diffusion dialysis with a Neosepta-AFN membrane is suitable for processing of this spent pickling solution.

### 4.2. Recovery and Rejection for Different Acid and Salt Concentrations

Furthermore, comparative measurements were performed at one volumetric liquid flow rate (150 mL h^−1^), where the initial hydrofluoric acid concentrations were changed from 2.0 to 4.0 mol L^−1^, while that of ferric nitrate were changed from 0.5 to 0.9 mol L^−1^.

For all these measurements, the rejection coefficients of Fe^3+^ ions were from 95.1% to 95.5%. Therefore, these values are practically independent of the acid and salt concentration in the feed, and they depend only on the volumetric liquid flow rate.

In Figure 4, recovery yields for the same salt concentration (0.7 M Fe(NO_3_)_3_) and different hydrofluoric acid concentrations are compared for the volumetric liquid flow rate of 150 mL h^−1^. It has been shown that recovery of nitrates and fluorides increases with the increasing acid concentration in the feed; on the contrary, the recovery of hydrons decreases. The recovery of nitrates is higher than that of fluorides, as shown above.

Figure 5 shows the comparison of the recovery yields for the same acid concentration (3 M HF) and different ferric nitrate concentrations for the volumetric liquid flow rate 150 mL h^−1^. Measurements with hydrofluoric acid without adding salt were also added to Figure 5. From the figure, it is evident that the recovery of nitrates and fluorides decreases with the increasing salt concentration in the feed; in contrast, the recovery of hydrons increases. The highest value of fluoride recovery (54%) was achieved for HF alone. Again, recovery of nitrates is higher than that of fluorides.

The actual composition of the spent pickling solution used was slightly different from the composition of model mixture which was chosen according to data from the literature [2]. The nitrate content in SPS was higher than that in MM, causing a decrease in the recovery of the nitrates (according to Figure 5). On the contrary, the fluoride content in SPS was lower than that in MM, which also caused a decrease in fluoride recovery (according to Figure 4).

## 5. Conclusions

Diffusion dialysis of model aqueous solutions of hydrofluoric acid and ferric nitrate was investigated in a two-compartment counter-current dialyzer with an anion-exchange membrane Neosepta-AFN. Furthermore, the dialysis of the real spent pickling solution that contains HF and Fe(NO_3_)_3_ as well as a small amount of Cr^3+^ and Ni^2+^ ions was studied. The basic transport characteristics, acid recovery yield and metal rejections, and their dependences on initial concentrations and volumetric liquid flow rates were evaluated from steady-state measurements. The permeability of the membrane and the separation factors were also determined. Experiments proved that the membrane is a good separator of ferric ions from the mixture of HF and HNO_3_ acids and that nitrate ions pass through the membrane significantly better than fluoride ions. Therefore, the resulting diffusate contains a significantly larger amount of nitric acid than hydrofluoric acid. It can be stated that continuous diffusion dialysis with a Neosepta-AFN membrane is suitable for the treatment of this spent pickling solution.

## Figures and Tables

**Figure 1 membranes-13-00009-f001:**
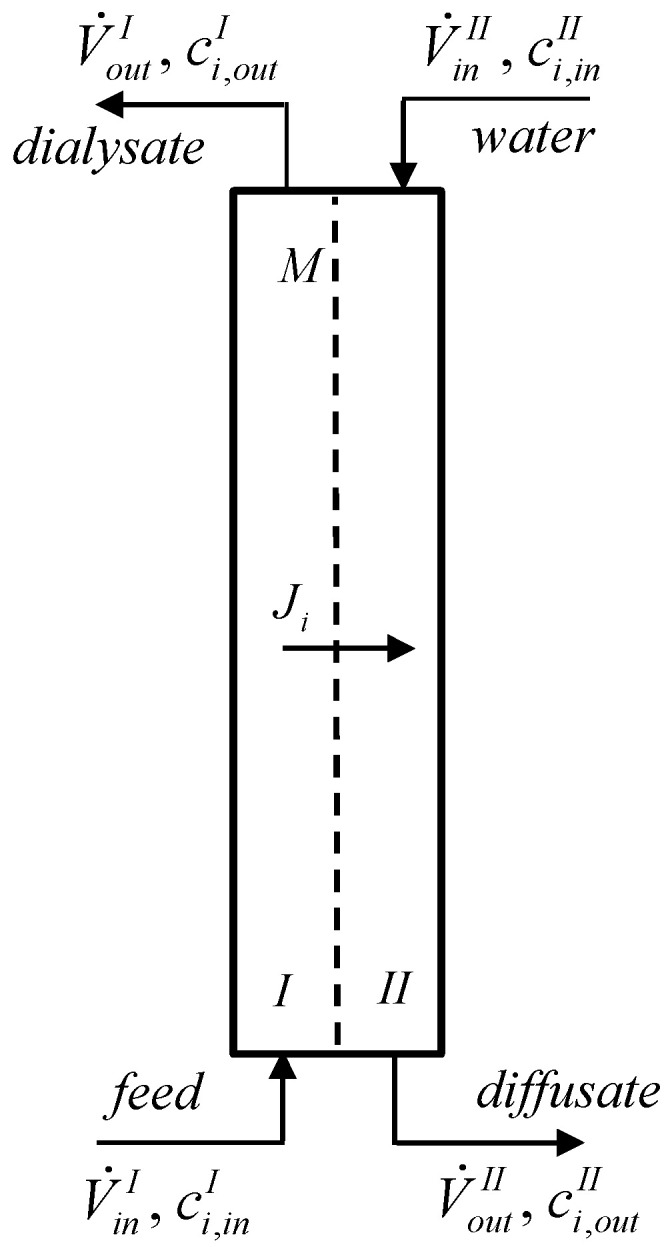
Scheme of the continuous dialyzer: *M*—membrane; *I* and *II*—compartments *I* and *II*.

**Figure 2 membranes-13-00009-f002:**
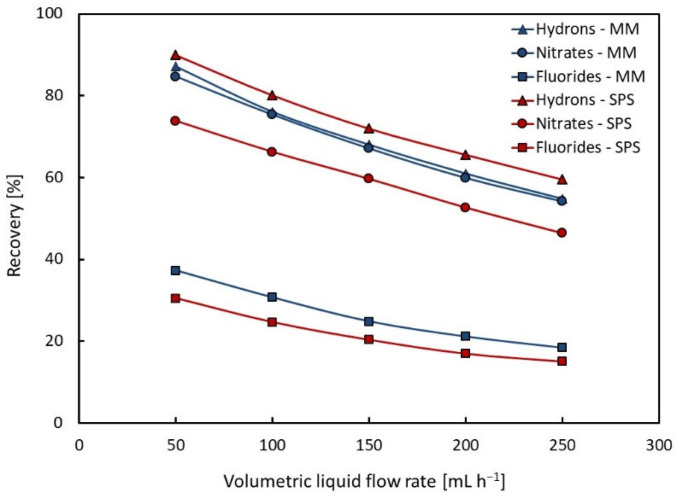
Dependence of recovery on volumetric liquid flow rate for model mixture (MM) and spent pickling solution (SPS).

**Figure 3 membranes-13-00009-f003:**
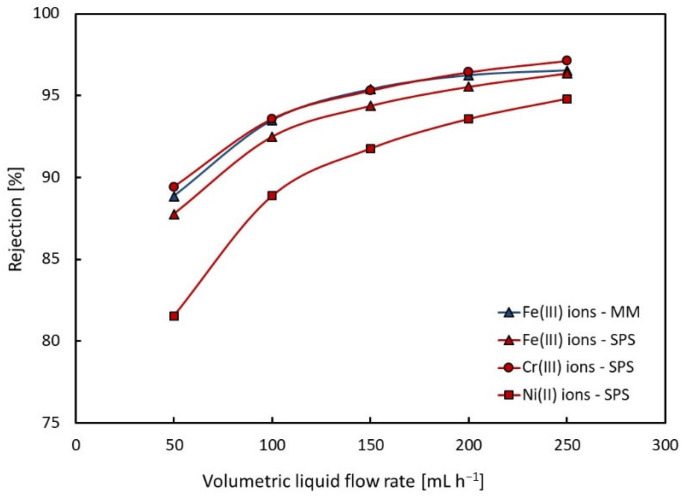
Dependence of rejection on volumetric liquid flow rate for model mixture (MM) and spent pickling solution (SPS).

**Figure 4 membranes-13-00009-f004:**
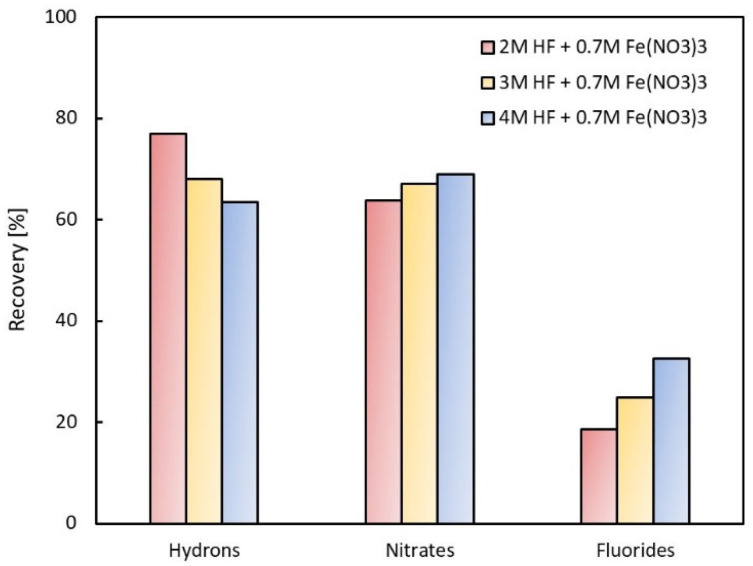
Comparison of recovery for different acid concentrations (150 mL h^−1^).

**Figure 5 membranes-13-00009-f005:**
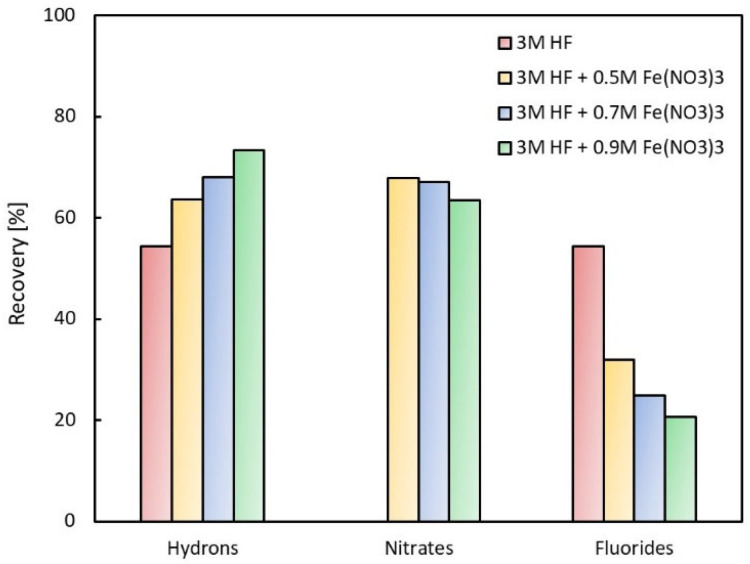
Comparison of recovery for different salt concentrations (150 mL h^−1^).

**Table 1 membranes-13-00009-t001:** The composition of tested solutions (mol L^−1^).

	Tested Model Mixture (3 M HF + 0.7 M Fe(NO_3_)_3_)	Tested Spent Pickling Solution	Typical Composition of Spent Pickling Solution [2]
NO_3_^−^ F^−^ H^+^ Fe^3+^ Cr^3+^ Ni^2+^	2.1	3.32	1.9–2.6
	3.0	2.52	3.2–4.2
	3.0 0.7 − −	3.04 0.68 0.21 0.089	2.6–4.0 0.6–0.8 0.1–0.2 0.05–0.1

**Table 2 membranes-13-00009-t002:** The composition of the dialysate and the diffusate (mol L^−1^) for 150 mL h^−1^.

	Dialysate		Diffusate	
	Model Mixture	Spent Pickling Solution	Model Mixture	Spent Pickling Solution
NO_3_^−^ F^−^ H^+^ Fe^3+^ Cr^3+^ Ni^2+^	0.70	1.51	1.37	1.95
	2.26	2.04	0.72	0.50
	0.98 0.68 − −	0.86 0.66 0.020 0.083	1.98 0.031 − −	2.15 0.038 0.0096 0.0072

**Table 3 membranes-13-00009-t003:** Recovery and rejection (for volumetric liquid flow rate 50–250 mL h^−1^).

		Model Mixture	Spent pickling Solution
	NO_3_^−^	54–85%	46–74%
Recovery	F^−^ H^+^	18–37% 55–87%	15–31% 60–90%
Rejection	Fe^3+^ Cr^3+^ Ni^2+^	89–97% − −	88–96% 89–97% 82–95%

**Table 4 membranes-13-00009-t004:** Permeability (*P_i_* × 10^6^ m/s) and separation factor.

		Model Mixture	Spent Pickling Solution
	NO_3_^−^	2.54	1.73
Permeability	F^−^ Fe^3+^	0.54 0.097	0.40 0.11
Separation factor	NO_3_^−^/Fe^3+^ F^−^/Fe^3+^	26.2 5.6	15.7 3.6

## Data Availability

The data presented in this study are available on request from the corresponding author.

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
