# Peer review of "Treatment of Spent Pickling Solutions by Diffusion Dialysis Using Anion-Exchange Membrane Neosepta-AFN"

_membranes, 2022, doi:10.3390/membranes13010009_

Round 1

Reviewer 1 Report

The authors of “Treatment of spent pickling solutions by diffusion dialysis using anion exchange membrane Neosepta-AFN” measured the permeabilities of Neosepta AFN to HF and Fe(NO3)3 and demonstrated Neosepta could be useful for pickling solution treatments. However, this reviewer believes a major revision is needed to meet the expectation of the readers.

General Comments:

65 - Can you explain more about the Neosepta AFN (i.e., styrenic AEM with quaternary ammonium group)?

Why did you select Neosepta AFN over other AEMs?

Have you considered different AEMs or charge-free RO membranes?

177 - How did you pre-treated the membrane?

If you did not, have you detected Chloride?

(or any sign of counterion exchange during the experiment?)

192 - It seems Nitrates are quite selective in Neosepta. Do you think the counterions (initially Cl-) would be exchanged to Nitrates (or F-) during the experiment?

If so, would that impact the results?

238 - Have you detected FeF3 on the feed side over time?

238 - Can you measure Fe(NO3)3 permeability by itself and compare the result with these results (with HF)?

If you believe this is the reason (line 233), I guess Fe(NO3)3 permeability by itself should be less than the one with HF.

256 - Any references to your statement 95 % rejection is high enough?

264 - Again, can you try 0.7M Fe(NO3)3 by itself?

Also, can you add the error bars for Figures 4 and 5?

Minor Comments: 

72 - write out CSTR

126 - this figure is blurry

162 - this subtitle is hard to read

Reviewer 2 Report

The authors present an interesting work in the field of diffusion dialysis. The use of the real solution is a good addition to the paper.

There is a number of comments which should be considered by the authors to further improve the quality of the paper.

1. Please, provide a reference for Eq. 3

2. Eq. 2 is written incorrectly if written in %. Usually rejection shows how much a matter is not transported through membrane. While in DD this characteristic is desirable for metal ions, it is inconvenient to use for acids (we want their rejection to be as close to zero as possible). Please add a description.

3. What is the value of dV/dz? How is it compared with ions flux (Ji)?

4. The flux of ions through ion-exchange membrane and adjacent boundary layers is determined by the limiting stage – either the diffusion through membrane or one of the layers. Please specify what the values of transport coefficients were. How they were established/measured?

5. Please, provide the width and height of each compartment. Was the DD module a spiral-wound module or a plate-and-frame module? Please be more specific in the description of the experimental installation.

6. Line 144 – remove closing parentesis “)”

7. Line 157 – H+ should be H+

8. Please, mark dialysate and diffusate streams on the fig. 1

9. Section 4.1.3 “The permeability of the membrane to individual ions was determined with the help of the procedure consisting of the numerical integration of Eqs. (3) and the one-dimensional optimization procedure described in Theory.” The Theory section does not include a description of the one-dimensional optimization procedure, but rather a reference to it. Please add the procedure description. In addition, I do not see how fitting of the experimental concentration vs time data can give values of three transport coefficients (Pi and two kL). It can only give either an integral value of permeability coefficient (which includes all three components) or the permeability of the limiting stage (the lowest value). Please specify which one you consider as permeability.

10. Given the data presented on figures 2-5 the optimization problem could be solved to obtain optimum parameters of the process. It would greatly improve practical value of the paper.

11. In the introduction section authors could cite a relevant reference https://doi.org/10.3390/membranes12121196

Round 2

Reviewer 1 Report

Please practice at least triplicate for transport values in the future.

Reviewer 2 Report

The authors have took into consideration my main concerns and in a good way. I think that the paper can be accepted in present form.